# Evaluation of Bleeding Resistance in Chip Seal and Asphalt Emulsion Residue Rheology

**Preeda Chaturabong**

Faculty of Engineering, King Mongkut's Institute of Technology Ladkrabang, Bangkok 10520, Thailand; preeda.ch@kmitl.ac.th

**Abstract:** Chip seal bleeding is influenced by many factors, including design inputs, material properties, and project-specific conditions. It reduces the surface texture of the pavement and thus compromises the safety of the traveling public. Even though factors that bring about premature bleeding are known, currently, no laboratory test methods for evaluating bleeding in chip seals have been specified. The objective of this paper is to present the results of an investigation of the influence factors of asphalt emulsion residue properties measured by the ASTM D7405 multiple stress creep and recovery (MSCR) test, as well as other factors related to chip seal bleeding resistance as measured by the modified loaded wheel test (MLWT). In this study, the MSCR test was used as a tool for evaluating the performance of asphalt emulsions because it has been identified as a potential test related to bleeding in the field. In addition, MLWT was selected as a tool for evaluating chip seal bleeding performance in the laboratory. The results of the MLWT showed that the emulsion application rate (EAR), aggregate gradation, and emulsion properties were significant factors affecting bleeding. The MSCR test was found to be a promising tool for the performance evaluation of asphalt emulsion residue, as the test was able to differentiate between emulsion chemistries and modifications in terms of sensitivity to both temperature and stress. In relation to chip seal bleeding resistance, only the creep compliance ($J_{nr}$) obtained from the MSCR test results was identified as a significant property affecting potential for bleeding.

**Keywords:** asphalt emulsion; chip seal; modified loaded wheel test; multiple stress creep and recovery; bleeding

## 1. Introduction

Chip seals are placed on a variety of roadways, including those with low and high traffic volumes, where low volume is defined as those with average daily traffic (ADT) less than 5000 and high volume is defined as those with more than 20,000 ADT. Polymer-modified asphalt emulsions are recommended for roads with a high traffic volume [1,2]. Also, inverted seals have successfully been used on high-traffic-volume (30,000 ADT) Australian pavements [3]. Since most of the traffic is composed of passenger cars and small trucks, it is important to select an appropriate range of stress levels for laboratory evaluation of bleeding.

Bleeding is an important failure, since it reduces the surface texture of the pavement and, hence, compromises the safety of the traveling public, particularly during wet seasons and at intersections. The bleeding performance of chip seals relies on many factors, including climatic conditions, traffic volume and type, aggregate properties, asphalt emulsion properties, emulsion application rate (EAR), and existing pavement surface. Aggregate properties, including size, shape, gradation, and toughness, influence bleeding performance.

Regarding material properties, both aggregates and asphalt emulsions contribute to bleeding resistance. For example, aggregate size and nonuniform aggregate gradations increase the potential for

bleeding; however, aggregate size can be accounted for by adjusting the EAR to provide an equivalent embedment percentage to smaller aggregates [4,5]. To resist bleeding, ideal asphalt emulsion residue properties include resistance to softening at increased temperatures and/or stresses. The performance of chip seals is largely dependent on the asphalt emulsion, as it is the binding component between the aggregates and the existing surface. Soft asphalt emulsion can result in bleeding in hot weather, since it can allow the movement of aggregates to the residue film on the existing surface while forcing the residue to move to the surface of the seal covering the aggregates. Therefore, stiffening an asphalt emulsion by modification could favorably influence the bleeding performance. However, since the cost of modified asphalt is higher than that of unmodified asphalt, the selection of a suitable asphalt emulsion for each location should be based on critical factors. Climate is one necessary factor that requires consideration regarding bleeding, because in hot weather, bleeding tends to occur more often due to the softening of the asphalt binder. This phenomenon allows chips to penetrate into the underlying binder, leaving excess asphalt binder on the surface [6]. Therefore, the asphalt emulsion type needs to be selected depending on the regional climate.

In addition to climate, the condition of the existing pavement surface and application rates also must be considered. Prior to selecting the target EAR, the existing pavement is surveyed, and an application rate correction factor is applied to account for the existing pavement surface condition. The application rate must be correct during construction in order to achieve optimum performance of the chip seal; if the emulsion application is excessively low, it will not retain chips in place under traffic and cause raveling, while if it is excessively high, bleeding in hot weather will occur, thus resulting in a loss of friction. Generally, the EARs are not considered as much with respect to the effect of repeated loadings for high traffic volumes [7]. This is because the heavy traffic will continue to embed the aggregates into the underlying surface after the road is open to traffic.

Bleeding generally occurs at high temperatures with high traffic stress and volume. Surface treatment specifications, while having been studied by many researchers, have not been finalized. Hanz et al. (2012) stated that the multiple stress creep and recovery (MSCR) test is capable of discriminating between asphalt emulsion type and the effect of asphalt emulsion modification; however, these efforts did not include comparisons to chip seal performance [8]. Kim et al. (2017) employed the MSCR test and $G^*/\sin\delta$ to assess the bleeding resistance of residual asphalt emulsions at high-temperature performance grades. It was found that the correlation between creep compliance ($J_{nr}$) values and high-temperature mixture performance for chip seals is good. Also, the MSCR test better captures the elastic response of the polymer network than the Dynamic Shear Rheometer (DSR) test employed to measure $G^*/\sin\delta$ [9].

In earlier research, the bleeding resistance of chip seal mixtures was evaluated through simulated loading using a third-scale model mobile loading simulator [10]. Although the results of the study are promising, the simulator equipment is costly and not widely available. Chaturabong et al. (2015) applied the modified loaded wheel test (MLWT) to evaluate bleeding in chip seals. This method was successful at measuring chip seal bleeding resistance because it eliminated raveling and better represented chip seals in the field [11].

In this study, we investigated the influence factors of asphalt emulsion residue properties, and other factors related to chip seal bleeding resistance were measured by the MLWT.

## 2. Materials and Testing Procedure

### 2.1. Materials

Five asphalt emulsions and two aggregate types were used in this study. The selected asphalt emulsions included cationic and anionic emulsions in order to account for the effects of emulsion chemistry, and polymer- and latex-modified asphalt emulsions to test for resistance to bleeding. The asphalt emulsions employed in this study were labeled CRS-2, CRS-2L, CRS-2P, HFRS-2, and HFRS-2P. All asphalt emulsions were combined with both granite and limestone aggregate

sources to study the effects of aggregate source properties. The properties of the aggregates used are given in Table 1. These properties met the specifications of AASHTO T96 [12] and ASTM D5821 [13] for percent Los Angeles (LA) abrasion and percent fractured faces accordingly.

**Table 1.** Aggregate types and properties used.

| Aggregate Properties | Specification | Granite | Limestone |
|---|---|---|---|
| % LA abrasion | Limit 35% | 18.1% | 23% |
| % Fracture Faces | 90–100 | 95% | 90% |

### 2.1.1. Aggregate Gradation

Chip seals are generally constructed with uniform gradations. The effects of gradation were evaluated by considering two gradations, namely, coarse and fine, based on the maximum aggregate size. The coarse gradation was made up of 50% passing the 9.5 mm size sieve and 0% passing the 6.5 mm size sieve. The fine gradation had 50% passing the 6.5 mm size sieve and 0% passing the 4.75 mm size sieve, as shown in Figure 1. In addition, all aggregates were washed to remove dust coatings from the larger particles.

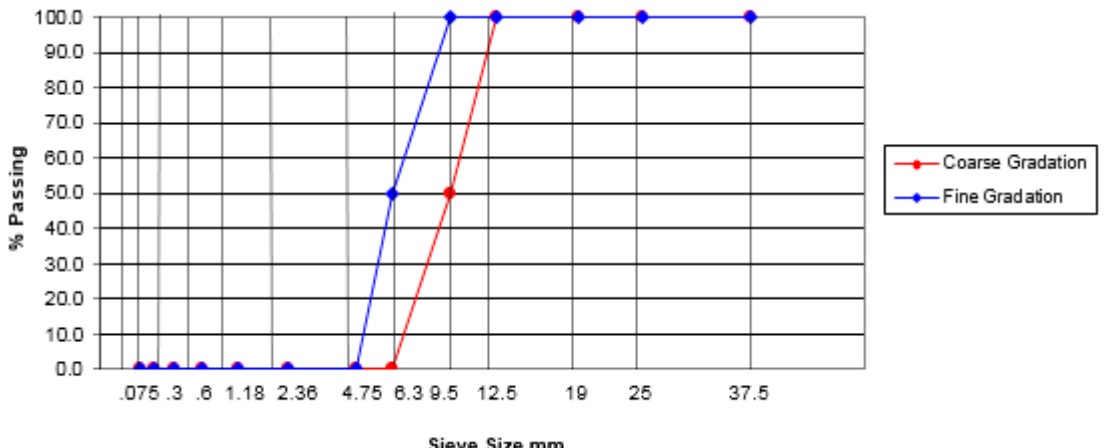

**Figure 1.** Gradation chart.

### 2.1.2. Material Application Rates

Material application rates for chip seal samples were determined using the equations of the modified McLeod's method [14]. In order to account the effect of EAR, two binder application rates of 35% and 70% air void filled were selected. Table 2 presents the material application rates used in this study.

**Table 2.** Emulsion and aggregate application rate.

| | | EAR (L/m$^2$) | | | |
|---|---|---|---|---|---|
| **% Void Filled** | **Aggregate Gradation** | **Asphalt Emulsion Type** | | | **Aggregate Application Rate (kg/m$^2$)** |
| | | **CRS-2** | **CRS-2L** | **CRS-2P** | |
| 35% | Fine | 1.02 | 0.95 | 0.97 | 8.13 |
| | Coarse | 1.51 | 1.42 | 1.44 | 11.89 |
| 70% | Fine | 2.03 | 1.91 | 1.93 | 8.13 |
| | Coarse | 3.02 | 2.84 | 2.87 | 11.89 |

## 2.2. Asphalt Emulsion Residue Performance Evaluation

The overall objective of the testing design was to investigate the influence factors of asphalt emulsion residue properties, as well as other factors related to chip seal bleeding resistance measured by the MLWT. The MSCR test, as specified in ASTM D 7405 [15] and AASHTO TP 70 [16] was used to evaluate if the $J_{nr}$ values correlate with bleeding resistance at high temperatures.

Test conditions were varied to assess both sensitivity to temperature and stress. AASHTO M 332 [17] provides specification criteria based on the $J_{nr}$ value for different traffic loading conditions. In the specification, the maximum allowable value for standard traffic (S) is equal to 4.5 kPa$^{-1}$; for subsequent traffic conditions associated with heavy (H), very heavy (V), and extremely heavy (E) traffic, the $J_{nr}$ threshold is reduced to adjust for increasing traffic levels. In this study, the range of test temperatures was selected to represent possible service temperatures experienced on roadways and to obtain $J_{nr}$ values above and below the S-grade threshold. To further assess stress sensitivity, a third stress level of 10 kPa was included in the test procedure. All asphalt emulsion residues were recovered using the low-temperature evaporative recovery procedure specified in AASHTO PP 72 [18]. This recovery method involves drawing the asphalt emulsion down to a film thickness of 381 μm on a silicone mat and curing it for 6 h at 60 °C in a forced draft oven.

## 2.3. Modified Loaded Wheel Test to Assess Chip Seal Bleeding

Previous research [8] has established that the MSCR test is capable of discriminating between asphalt emulsion type and the effect of asphalt emulsion modification; however, these efforts did not include comparisons to chip seal performance. As a result, a test is needed to validate that the differences observed in asphalt emulsion residue $J_{nr}$ values are related to chip seal performance and to apply this relationship to propose performance limits for bleeding.

In this study, applying the MLWT to evaluate bleeding in chip seal modifications was done using a standard device that included the addition of a temperature control device, a base plate to secure the chip seal sample, a rubber mat on the top of the sample to protect tire wear, and neoprene foam under the sample to represent existing flexible pavement. This method, which eliminates raveling and better represents chip seals in the field, is capable of measuring chip seal bleeding resistance [11]. The clamps used to hold the sample in place were also modified, as shown in Figure 2.

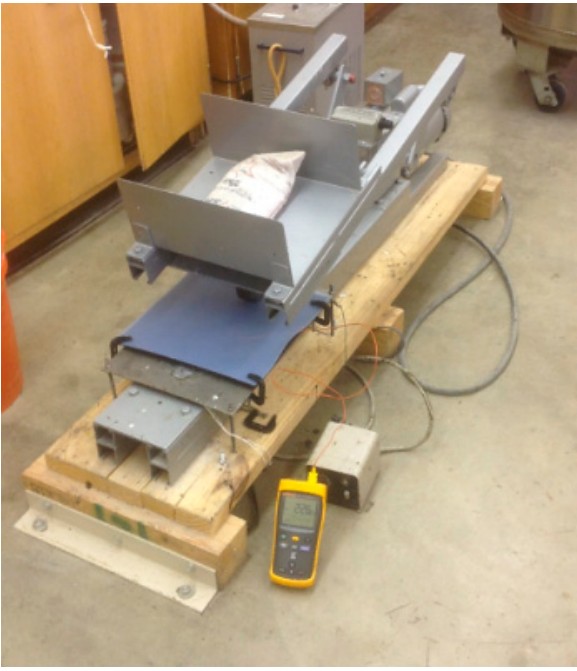

**Figure 2.** Diagram of the modified loaded wheel test (MLWT).

Samples consisting of asphalt emulsion/aggregate combinations were prepared according to the guidance provided in ASTM D7000 [19]. After curing, samples were placed on the MLWT support and heated to the specified testing temperature. Testing included two replicates for each sample by rotating the sample 90° to the wheel path. Once the test was completed, the sample was scanned, and a digital image was taken and processed to compare the initial image with the same after trafficking. To conduct this analysis, IPAS² software, developed by UW-Madison, was used to convert the captured image to a binary (black and white) image using well-established image threshold techniques [20]. An example of the application of image analysis to quantify bleeding is provided in Figure 3.

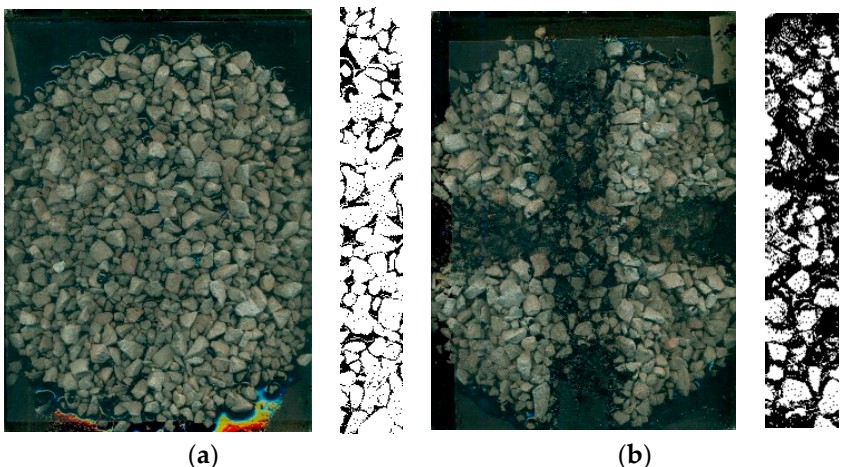

(**a**)                                                (**b**)

**Figure 3.** Prepared sample and resulting black-and-white image (**a**) prior to bleeding test and (**b**) after bleeding test.

The extent of bleeding was then calculated using Equation (1):

$$Bleeding\ (\%) = [(AAinitial - AA_x)/AA_{initial}] \times 100 \tag{1}$$

where

$AA_x$ = area fraction of aggregate (white color) after x cycles, and

$AA_{initial}$ = area fraction of aggregate (white color) prior to testing.

Since both images were processed using the same filtering threshold, the difference could be assumed to be the effect of bleeding. The smaller the white area (aggregates) after x cycles, the greater the bleeding percent.

*2.4. Experimental Design*

The experimental plan is given in Table 3. This experiment was carried out with granite aggregate at a constant stress level of 299 kPa (43.4 psi) for 1 day of curing with the loading cycle kept at 200 cycles. The test procedure used to evaluate residue properties is described in subsequent sections.

In the experimental design, there were three asphalt emulsion types and one aggregate type for conducting the test to quantify the factors influencing bleeding of chip seals. To verify if the significant factors from the main experimental design are valid, an additional "null" experiment was conducted to verify that the measured and estimated bleeding based on the model derived from the main experiment was well correlated. For the null experiment, different asphalt emulsions and aggregates from those used in the main experiment were selected. Specifically, the null experiment used limestone coarse gradation with a stress of 299 kPa (43.4 psi), 1 day of curing time, and 200 loading cycles. The null experiment matrix is shown in Table 4.

**Table 3.** Experimental design for assessing the effects of design factors on bleeding resistance and the relationship between creep compliance ($J_{nr}$) and bleeding resistance with the MLWT.

| Factors | Levels | Description | Parameter Measured |
|---|---|---|---|
| Asphalt Emulsion Type | 3 | CRS-2, CRS-2P, and CRS-2L | – |
| Aggregate Gradation | 2 | Fine and coarse | – |
| Emulsion Application Rate (EAR) | 2 | 30% and 70% of voids filled | % Area with bleeding (by imaging) |
| Temperature | 3 | $J_{nr}$@ 3.2 kPa—0.5, 2.0, and 5.0 (adjust temperature to keep $J_{nr}$ constant) | – |
| Replicate | 2 | 2 | – |
| Total Tests | 72 | – | – |

**Table 4.** Null experiment design for verifying the best subsets for bleeding resistance with the MLWT.

| Factors | Levels | Description | Parameter Measured |
|---|---|---|---|
| Asphalt Emulsion | 2 | HFRS-2 and HFRS-2P | |
| EAR | 2 | %void filled = 70% | % Area with bleeding |
| Temperature | 2 | $J_{nr}$@3.2 kPa—2.0 and 5.0 | |
| Replicate | 2 | 2 | – |
| Total Tests | 16 | – | – |

As shown in Table 3, three levels of $J_{nr}$ (0.5, 2, and 5 kPa$^{-1}$) were selected to represent low, medium, and high bleeding potential, respectively. Testing was carried out on different asphalt emulsion residues to determine the test temperature at which the $J_{nr}$ at the stress level of 3.2 kPa was equal to 0.5, 2.0, and 5 kPa$^{-1}$. The temperatures determined for different emulsions are presented in Table 5.

**Table 5.** Test temperatures at which different asphalt emulsions met the specified $J_{nr}$ values for a stress level of 3.2 kPa.

| Asphalt Emulsion | $J_{nr}$ at 3.2 kPa | Temperature (°C) |
|---|---|---|
| CRS-2 | | 45 |
| CRS-2L | 0.5 | 46 |
| CRS-2P | | 53 |
| CRS-2 | | 51 |
| CRS-2L | 2.0 | 53 |
| CRS-2P | | 60 |
| CRS-2 | | 59.5 |
| CRS-2L | 5.0 | 61 |
| CRS-2P | | 67 |

## 3. Experimental Results

Chip seal samples were prepared and tested with the MLWT at the test temperatures corresponding to asphalt emulsion $J_{nr}$ values of 0.5, 2, and 5 kPa$^{-1}$, as listed in Table 5. The results for the EAR of 35% air void filled are presented in Figure 4 for the fine and coarse gradations. The results for both gradations show a positive relationship between $J_{nr}$ and percent bleeding of chip seals. As the $J_{nr}$ increased, the amount of bleeding was also observed to increase for all three asphalt emulsions and both aggregate gradations.

The percent bleeding almost doubled when the $J_{nr}$ increased from 0.5 to 5 kPa$^{-1}$. This indicates that a $J_{nr}$ at 3.2 kPa could be a good indicator of potential asphalt emulsion bleeding. The results, however, showed minimal sensitivity to asphalt emulsion type, irrespective of aggregate gradation, except for the $J_{nr}$ equal 5 kPa$^{-1}$ for the fine gradation. This implies that, under these test conditions, latex- and polymer-modified asphalt emulsions, as well as unmodified asphalt emulsions, will provide similar resistance to bleeding if they have equal $J_{nr}$ values at the testing temperature.

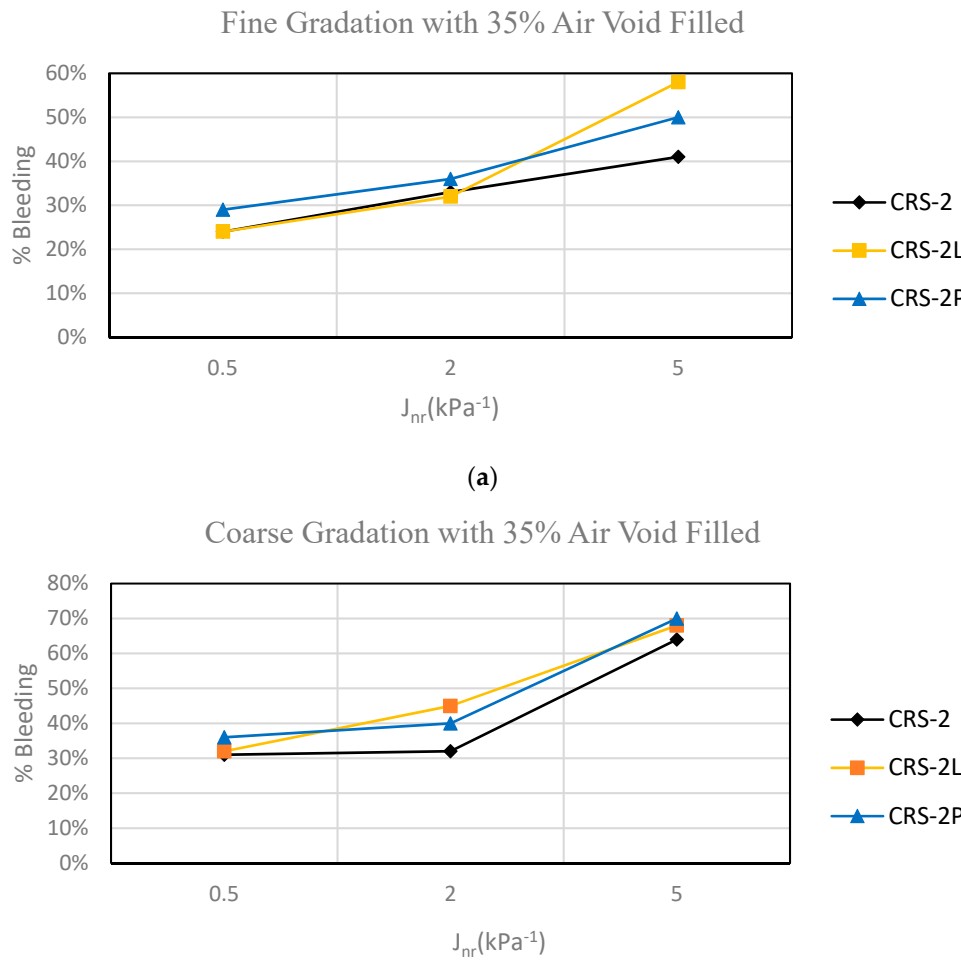

**Figure 4.** Relationship between $J_{nr}$ and percent chip seal bleeding for (**a**) fine gradation with low EAR and (**b**) coarse gradation with low EAR.

The results for the high EAR (70% air voids filled) are shown in Figure 5 for the fine and coarse gradations. Similar to the trend observed for the low EAR, a positive relationship between percent bleeding on chip seals and $J_{nr}$ can be noted. Percent bleeding also doubled when the $J_{nr}$ increased from 0.5 to 5 kPa$^{-1}$, indicating that a residue with a high $J_{nr}$ may be prone to bleeding. For fine gradation, the percent bleeding ranged between 30% and 38%, 41% and 49%, and 59% and 65% for the $J_{nr}$ values of 0.5, 2.0, and 5.0 kPa$^{-1}$, respectively, for all three asphalt emulsion types. Similarly, the percent bleeding for coarse gradation increased at higher $J_{nr}$ values. The difference of percent bleeding between fine and coarse gradations for the high emulsion rate varied depending on the $J_{nr}$ value. The percent bleeding values for coarse gradation were greater than those for fine aggregates by 17%, 24%, and 18% for $J_{nr}$ values of 0.5, 2.0, and 5 kPa$^{-1}$, respectively.

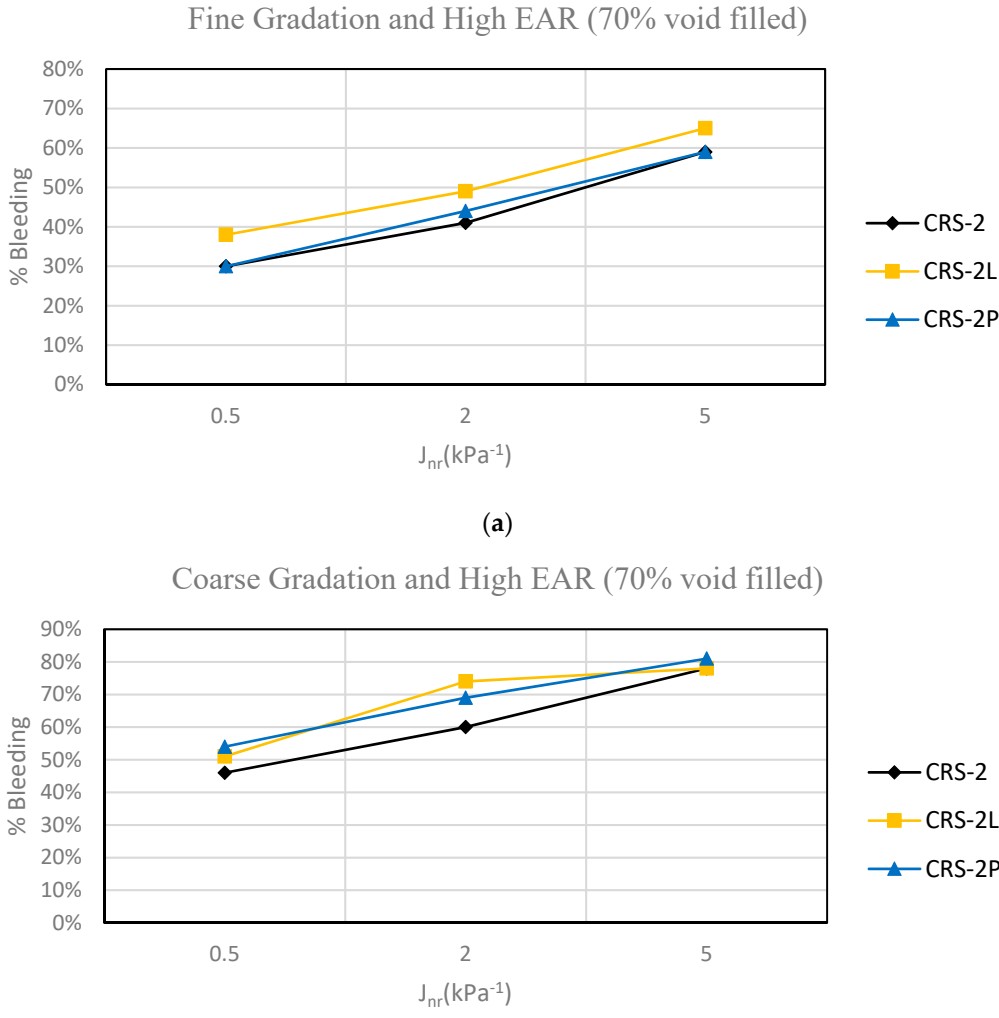

(**a**)

(**b**)

**Figure 5.** Relationship between $J_{nr}$ and percent chip seal bleeding for (**a**) fine gradation with low EAR and (**b**) coarse gradation with high EAR.

The results for the high EAR also showed less sensitivity to asphalt emulsion modification types (P and L) than to $J_{nr}$ values. There was a noticeable difference between modified and unmodified asphalt emulsions in terms of percent bleeding at all values of $J_{nr}$. These results indicate that $J_{nr}$ is related to percent bleeding for both conventional and modified asphalt emulsion types and that the effect of modification is marginal when compared at the same $J_{nr}$ value. Recall that the experiment controlled the $J_{nr}$ value rather than the temperature in the MLWT comparisons. As a result, modified asphalt emulsions achieved a $J_{nr}$ value of 5.0 kPa$^{-1}$ at temperatures of 1.5–6.0 °C higher than the control.

Furthermore, the results showed that coarse gradation provided more percent bleeding than fine aggregates in all conditions. Since the aggregate shape of the coarse aggregates was more angular, this resulted in nonuniformity of the chip spread. This can lead to more stress concentration on the contact area, and this assumption can be verified by the results shown above.

Figure 6 shows a composite summary of previously presented results. The labels for this figure indicate emulsion–gradation (F is fine, C is Coarse) $J_{nr}$ at 3.2 kPa. All high EARs showed higher percent bleeding than low EARs. Chip seals with low EARs had percent bleeding in the range of 21%–70%, while chip seals with high emulsion rates had percent bleeding in the range of 30%–81%.

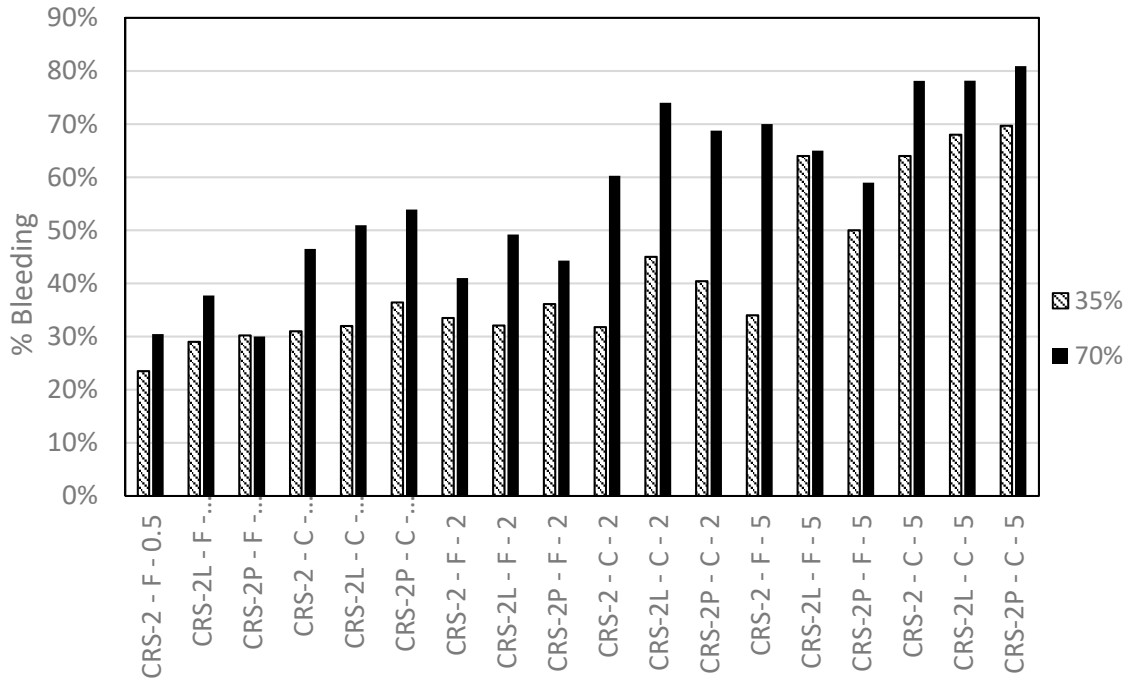

**Figure 6.** Effect of EAR on bleeding resistance.

*3.1. Factors that Influence Bleeding Resistance*

The information presented in the previous figures identify many factors that have a potential impact on bleeding resistance, including aggregate gradation, EAR, and $J_{nr}$ at 3.2 kPa. Statistical analysis was used to quantify the significance of these factors and assess their relative contribution to bleeding resistance. Prior to conducting the analysis, factors were added or modified to best represent the materials used. The material properties considered for the analysis included those obtained from MSCR testing (percent recovery at 3.2 kPa). Moreover, aggregate gradation for statistical analysis was quantified by fitting the gradation curve to a cumulative Weibull distribution; parameters $\kappa$ and $\lambda$ denote the shape (fineness or coarseness) and the scale (dense or open/gap-graded) of aggregate type, respectively.

The Weibull distribution was fitted to the gradation curves in order to calculate the gradation shape parameters. The parameters calculated were $\kappa$ and $\lambda$, which denote the shape (fineness or coarseness) and the scale (dense or open/gap-graded) of aggregate type, respectively. These parameters were determined using Equation (2). These parameters were used in the statistical analysis to evaluate the effects of gradation:

$$P(x) = 1 - e^{-\left(\frac{x}{\lambda}\right)^{\kappa}} \tag{2}$$

where

P(x) = percent finer than sieve x,

x = sieve size in millimeters to the 0.45 power, and

$\lambda$, $\kappa$ = curve fit parameters of shape and scale, respectively.

Analysis of variance (ANOVA) was conducted at a significance level of 0.1 to screen the significant factors affecting bleeding. For the analysis, the software used was R-project [21]. The definitions of each factor are as follows: Gradation_ $\lambda$: Weibull distribution parameter which denotes the shape (fineness or coarseness), where a high $\lambda$ value means greater coarseness; EAR: EAR; $J_{nr}$_3.2: $J_{nr}$ at 3.2 kPa; R_3.2: percent recovery at 3.2 kPa; Rep: replicates. The ANOVA results are shown in Table 6.

**Table 6.** ANOVA table for bleeding test.

| Variables | F-Value | Pr(>F) | Significance |
|---|---|---|---|
| Gradation_ $\lambda$ | 108.43 | <0.001 | *** |
| EAR | 104.00 | <0.001 | *** |
| $J_{nr}$_3.2 | 139.74 | <0.001 | *** |
| R_3.2 | 2.97 | 0.090 | . |
| Rep | 0.51 | 0.479 | |

Note: Significance codes—0 '***' 0.001 '**' 0.01 '*' 0.05 '.' 0.1 ' ' 1.

The results presented in Table 6 indicate that Gradation_ $\lambda$, EAR, $J_{nr}$_3.2, and R_3.2 were statistically significant factors affecting bleeding resistance. The significant factors can be ordered by the F-value, and the F-values of $J_{nr}$_3.2, Gradation_ $\lambda$, EAR, and R_3.2 are the ordered significant factors related to bleeding resistance. Moreover, the results show that the replicate was not significant for the bleeding resistance, indicating that the results are reliable and the test method is repeatable.

*3.2. Best Subsets Regression*

As the study selected the constant traffic stress, volume, and existing pavement, the factors emphasized included asphalt emulsion rheology, EAR, and aggregate gradation. Therefore, the measured bleeding was dependent on five factors, as stated in the previous section. However, to design the model for bleeding resistance, only four factors were considered when conducting the regression analysis. A best subsets regression was used to identify factors to include in a prediction model. The independent variables considered in this analysis included Gradation_ $\lambda$, EAR, $J_{nr}$_3.2, and R_3.2.

Table 7 shows the results of the best subsets analysis performed in the statistical analysis program Minitab. The methodology used to choose the best subsets was based on a high $R^2_{adj}$ value and the close value of a low Mallows's Cp and the number of variables in the chosen subset.

**Table 7.** Factor selection by best subset regression.

| Variables | $R^2_{adj}$ | Mallows's Cp | Gradation_ $\lambda$ | EAR | $J_{nr}$_3.2 | R_3.2 |
|---|---|---|---|---|---|---|
| 1 | 49.2 | 272.4 | | | X | |
| 2 | 68.5 | 142.3 | X | | X | |
| 2 | 67.7 | 147.7 | | X | X | |
| 3 | 87.5 | 17.7 | X | X | X | |

Using the subset outlined in Table 7, a quantitative prediction model was defined, as shown in Equation (3):

$$\% \text{ bleeding} = -0.124 + 0.0316 \text{ Gradation\_} \lambda + 0.410 \text{ EAR} + 0.0628 \text{ J}_{nr}\_3.2, \ R^2_{adj} = 87.5\% \quad (3)$$

where

% Bleeding = Estimated percent coated aggregates by total area,

Gradation_ $\lambda$ = Weibull distribution parameter describing the shape (fineness or coarseness),

EAR = Emulsion application rate, and

$J_{nr}$_3.2 = $J_{nr}$ at 3.2 kPa.

The model in Equation (3) was used to select the optimum asphalt emulsion, EAR, and aggregate gradation considering the constant existing pavement, traffic volume, traffic stress, and construction quality. In the best subset, the parameters which were included in the regression equation were Gradation_$\lambda$, EAR, and $J_{nr}$_3.2. There was no need to include percent recovery since there was a high

correlation between $J_{nr}$ and percent recovery. Also, it would be very difficult to control $J_{nr}$ and percent recovery independently.

The selected model (Equation (3)) showed that bleeding is high when Gradation_$\lambda$ is coarse (high) and EAR and $J_{nr}$_3.2 are high. The coefficients for each factor showed that the most significant factor to cause bleeding is EAR, followed by $J_{nr}$_3.2 and Gradation_$\lambda$. The best way to use this equation is to input the value of the required EAR and the gradation ($\lambda$) value and select the maximum value of $J_{nr}$ at the specific climate conditions (pavement temperature) that will lead to the maximum percent bleeding allowed. The equation indicates that finer aggregate gradation and lower EAR are also favorable.

As stated earlier, to verify the equation for the bleeding resistance, a new (null) experiment needed to be carried out in the MLWT. For the null experiment, the asphalt emulsions and aggregate selected were different from the main experiment. The aggregate for this experiment was limestone with 23% LA abrasion and 90% fractured face.

The results shown in Figure 7 indicate that the values from the equation were consistent with the value from the imaging analysis. The label in the plot indicates asphalt emulsion, $J_{nr}$ at 3.2 kPa, and EAR. This indicates that all factors in the regression analysis significantly affected bleeding resistance.

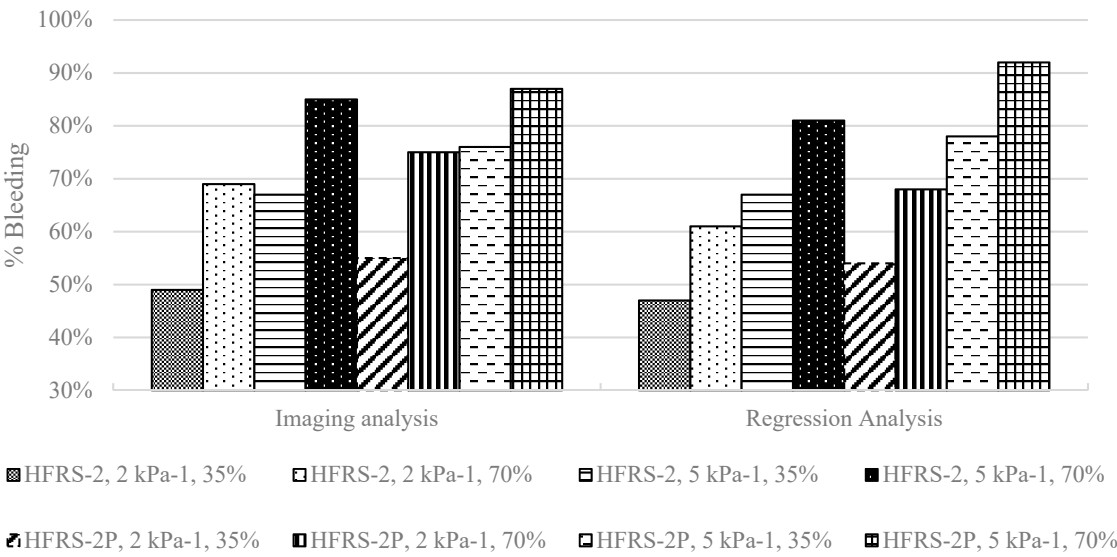

**Figure 7.** Comparison of measured percent bleeding (imaging analysis) and estimated percent bleeding (regression analysis).

## 4. Findings and Conclusions

This study evaluated the relationship between asphalt emulsion residue $J_{nr}$ values measured with the MSCR standard protocol and chip seal bleeding measured in the laboratory using the MLWT device. The following points summarize the findings:

1.  The MSCR test is a promising evaluation tool for determining the effect of asphalt emulsion residue properties on the bleeding performance of chip seals. It is clear that a $J_{nr}$ at 3.2 kPa value can differentiate between the effects of asphalt emulsions in terms of sensitivity to bleeding under different temperatures and cycles. As a result, it has the potential to be used to identify materials more prone to bleeding due to softening related to temperature.
2.  The results indicate that the EAR, aggregate gradation, and $J_{nr}$ at 3.2 kPa contribute to the bleeding of chip seals. As expected, the highest impact observed was for the EAR when it was changed from 35% to 70%. However, for each EAR, the $J_{nr}$ value of the asphalt emulsion residue at the test temperature had the second largest effect. Thus, the laboratory results indicate that for a given EAR, bleeding performance can be controlled through proper evaluation of asphalt emulsions at

the project climate using MSCR testing. However, the existing pavement, traffic volume, traffic stress, and construction quality need to be taken into consideration for future work.

3. Asphalt emulsions modified with polymers have lower $J_{nr}$ values and thus have less of a tendency to allow bleeding at the same temperature.

4. MSCR $J_{nr}$ at a 3.2 kPa value is a property of asphalt emulsion residue that is only one of the factors influencing bleeding resistance measured by MLWT.

5. Temperature control can be improved by creating a chamber for controlling the temperature.

6. The limitation of this study is that only a single chip seal was evaluated for bleeding. Many different multilayer seal treatments are currently used in practice, so further research is needed to evaluate if the concepts presented here can be applied to other seal systems.

7. Future research with additional technologies (i.e., emulsions and traditional testing) is needed for improving the validity of the testing.

**Funding:** This research received no external funding.

**Conflicts of Interest:** No conflict of interest.

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
