# Peer review of "Evaluation of Bleeding Resistance in Chip Seal and Asphalt Emulsion Residue Rheology"

_coatings, doi:10.3390/coatings9100670_

Round 1

Reviewer 1 Report

This is reviewers report on the manuscript entitled: 'Evaluation of Bleeding Resistance in Chip Seal and 2 Asphalt Emulsion Residue Rheology'.

The paper is well written, author has carried out and presented a substantial amount of work. However, the manuscript lacks focus and authors fail to draw out conclusions based on the amount of work that they have presented in the work presented. Manuscript reads like a section of a dissertation, it needs to be improved.

My recommendations to the author are the following:

Abstract needs to be rewritten that will address What, Why and How work is conducted as it is very confusing.  The author must define and follow through what is key objective of the study. Objective given in abstract and section 2.2 ln. 93 -94 do not match. ln 167: EAR (EAR) - Remove EAR in brackets  Use the same gradation in the graph figures (X and Y-axis), especially when comparing results, e.g. figure 4 a) and b) Pg. 9 ln 201 -202: statement not clear, please explain  ln 208: course gradation gives more % bleeding than fine gradation - this is expected because fine gradation has higher Surface Area Factor. I don't see the point of the statistical analysis? I would like to ask the author to explain the purpose of the statistical analysis and if not relevant please remove it.  

Author Response

Reviewer 1

Thank you very much for your review.

The paper is well written, author has carried out and presented a substantial amount of work. However, the manuscript lacks focus and authors fail to draw out conclusions based on the amount of work that they have presented in the work presented. Manuscript reads like a section of a dissertation, it needs to be improved.

Please find revised manuscripts per your suggestions.

My recommendations to the author are the following:

Abstract needs to be rewritten that will address What, Why and How work is conducted as it is very confusing. 

Please find the revised abstract.

The author must define and follow through what is key objective of the study. Objective given in abstract and section 2.2 ln. 93 -94 do not match.

Please find the revised the objective conformity. I already have the objective conformed.

 ln 167: EAR (EAR) - Remove EAR in brackets 

Please find revised in manuscript.

Use the same gradation in the graph figures (X and Y-axis), especially when comparing results, e.g. figure 4 a) and b)

I would like to show that with the same EAR and different gradation, gradation is an influence factor to bleeding resistance by using MLWT.

Pg. 9 ln 201 -202: statement not clear, please explain 

Please find revised manuscript. “To verify if the significant factors from the main experimental design are valid, an additional “null” experiment was conducted….”

ln 208: course gradation gives more % bleeding than fine gradation - this is expected because fine gradation has higher Surface Area Factor.

As per the result from the MLWT, it is conformed the expectation.

I don't see the point of the statistical analysis? I would like to ask the author to explain the purpose of the statistical analysis and if not relevant please remove it.  

The statistical analysis was added to show what factors statistically influence bleeding of chip seals so that the designer can adjust the chip seal design based on the statistical factors and equation.

Reviewer 2 Report

Overall good work. It is recommended to check the references and add more references to the background section.

Also, the experimental design is well organized but very limited and conclusions may not be applicable to a much larger experimental design. It is important to acknowledge limitations of the study.

Author Response

Reviewer 2

Thank you very much for your kind review.

Overall good work. It is recommended to check the references and add more references to the background section.

I added more references and amended the abstract and background to be better understanding.

Also, the experimental design is well organized but very limited and conclusions may not be applicable to a much larger experimental design. It is important to acknowledge limitations of the study.

Please find revised conclusions and limitations per your suggestion in manuscript. The MSCR test is a promising evaluation tool for effect of asphalt emulsion residue properties on bleeding performance of chip seals. It is clear that Jnr2 value can differentiate between effects of asphalt emulsions in terms of sensitivity to bleeding under different temperatures and cycles. As a result, it has potential to be used to identify materials more prone to bleeding due to softening related to temperature.

Results presented indicate that the EAR (EAR), aggregate gradation, Jnr at 3.2kPa contribute to the bleeding of chip seals. As expected, the highest impact observed is for EAR when changed from 35% to 70%, however for each EAR, the Jnr value of the asphalt emulsion residue at the test temperature has the second largest effect. Thus, laboratory results indicate that for a given EAR, bleeding performance can be controlled through proper evaluation of asphalt emulsions at project climate using MSCR testing. However, the existing pavement, traffic volume, traffic stress, construction quality need to take into consideration for the future work.

Asphalt emulsions modified with polymer have lower Jnr values and thus have less tendency to allow bleeding at the same temperature.

The temperature controlling can be improved by creating a chamber for controlling the temperature.

The limitation of this study is that only the single chip seal was evaluated the bleeding. It is understood that many different multi-layer seal treatments are currently used in practice, further research is needed to evaluate if the concepts presented can be applied to other seal systems.
